# Elevated plasma progranulin levels in the acute phase are correlated with recovery of left ventricular function in the chronic phase in patients with acute myocardial infarction

**Shingo Minatoguchi[1], Atsushi Satake[2], Hirotaka Murase[2], Ryo Yoshizumi[2], Hisaaki Komaki[2], Shinya Baba[2], Shinji Yasuda[2], Shinsuke Ojio[2], Toshiki Tanaka[1], Hiroyuki Okura[1], Shinya Minatoguchi[2]***

**1** Department of Cardiology, Gifu University Graduate School of Medicine, Gifu, Japan, **2** Department of Cardiology, Gifu Municipal Hospital, Gifu, Japan

* minatoguchi.shinya.g7@a.gifu-u.ac.jp

## Abstract

### Background

Progranulin is a secreted glycoprotein that regulates inflammation and wound healing. However, plasma progranulin levels in the acute phase and their clinical significance in patients with acute myocardial infarction (AMI) remain to be elucidated.

### Objective

We aimed to investigate the relationship between the increase in plasma progranulin levels in the acute phase and the recovery of left ventricular function in the chronic phase in AMI patients.

### Method and result

Eighteen AMI patients were followed up for 6 months. Blood samples were collected from the antecubital vein on day 0 (on admission) and day 7 in the acute phase. The control group consisted of patients without significant coronary artery stenosis, as assessed by cardiac catheterization (n = 16). Plasma progranulin levels were measured by enzyme-linked immunosorbent assay. Echocardiography was performed in the acute (within 7 days) and chronic (6 months) phases of AMI to evaluate left ventricular ejection fraction using the modified Simpson's method. Plasma progranulin levels in the AMI group on day 0 (69.5 ± 24.6 ng/mL) were similar to those in the control group (84.2 ± 47.1 ng/mL). There was a significant increase in progranulin levels in the AMI group on day 7 (104.2 ± 52.0 ng/mL) compared with day 0. The increase in plasma progranulin levels in the acute phase was positively correlated with the increase in left ventricular ejection fraction between the acute and chronic phases. Among various factors, only plasma progranulin levels were favorably correlated with left ventricular functional recovery in the chronic phase.

**Data Availability Statement:** All relevant data are within the manuscript and its Supporting Information files.

**Funding:** The author(s) received no specific funding for this work.

**Competing interests:** The authors have declared that no competing interests exist.

## Conclusion

The increase in plasma progranulin levels in the acute phase may serve as a predictive biomarker and a contributer for the recovery of left ventricular function in the chronic phase in patients with AMI.

## Introduction

Progranulin is a secreted glycoprotein that is present in macrophages, neutrophils, adipocytes, and skeletal myocytes [1,2]. Progranulin regulates inflammation [3,4] and wound healing [5], and is associated with diseases including frontotemporal dementia [6], rheumatoid arthritis [7], and cancer [8]. It was previously reported that the administration of recombinant progranulin attenuated neuronal injury by inhibiting neutrophil recruitment in a focal cerebral ischemia-reperfusion injury murine model [9]. A previous study reported that the intravenous administration of progranulin reduces myocardial infarct size and improves left ventricular (LV) function through reducing the inflammation of the damaged heart in murine and rabbit models of acute myocardial infarction (AMI) [10]. However, plasma progranulin levels in the acute phase and their clinical significance in patients with AMI remain to be elucidated. In the present study, we measured changes in plasma progranulin levels in the acute phase and investigated their correlation with LV function in the chronic phase at 6 months after the onset of AMI.

## Subjects and methods

This was a prospective cohort study. Inclusion criteria were: patients with AMI; admitted to Gifu Municipal Hospital or Gifu University Hospital because of anterior chest pain or discomfort; underwent coronary angiography; and were diagnosed as AMI based on the presence of prolonged anterior chest pain, ST segment elevation in electrocardiogram, and an occluded coronary artery by coronary angiography. AMI patients (n = 18; 14 males and 4 females) were treated with percutaneous coronary intervention (PCI) followed by standard pharmacological treatment. The mean age of AMI patients was 69.2 ± 14.1 years old. The control group (n = 16; 10 males and 6 females) consisted of patients without significant coronary artery stenosis who underwent cardiac catheterization because of precordial complaints. All the AMI patients in the present study were performed complete revascularization of the occluded coronary artery, which has been reported to reduce the risk of heart failure hospitalization and cardiovascular death [11]. Some of the control group had been previously treated with percutaneous coronary intervention because of coronary stenosis and administered drugs. The mean age of the control was 76.8 ± 8.5 years old. We started the recruitment of the patients on the 1st of May 2019 and ended on 31th of March 2022. Allocation of the patients is shown in Fig 1. The clinical study was performed according to the CONSORT Guidelines.

This study was approved by the Ethics Committee of Gifu Municipal Hospital (approval numbers: 455). Written informed consent was provided from all patients before the study commenced. The study conformed with the principles outlined in the Declaration of Helsinki (Br Med J 1964; ii:177). Public registry and trial registry number was UMIN000040165.

### Measurement of plasma progranulin levels

For the measurement of plasma progranulin levels, blood samples were collected from the antecubital vein on day 0 (on admission) and day 7 in the acute phase in AMI patients and the

# Participants flow diagram

**Anterior chest complaints (n=34)**

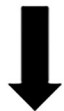

**Coronary angiography (n=34)**

**Acute myocardial infarction**

**Without significant coronary artery stenosis**

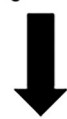

**AMI group (n=18)**

**Control group (n=16)**

**Fig 1. Allocation of the patients into AMI group and control group.**

control. The samples were collected into sterile tubes containing EDTA, immediately placed on ice, centrifuged at 1,500 g for 10 min at 4°C, and rapidly frozen and stored at -80°C until analysis. Plasma progranulin levels were measured by enzyme-linked immunosorbent assay (ELISA) (Progranulin [human] ELISA Kit, AdipoGen, Switzerland), according to the manufacturer's instructions. The repeatability of plasma progranulin level measurements was confirmed by assessing the same samples multiple times (average coefficient of variation less than 0.03, n = 5).

## Measurement of cardiac function by echocardiography

Cardiac echocardiography (iE33, PHILIPS, Tokyo) was performed during the acute (within 7 days) and chronic (6 months) phases of AMI. Left ventricular ejection fraction (LVEF), LV end-systolic dimension (LVSd), and LV end-diastolic dimension (LVDd) were measured. LVEF was assessed using the modified Simpson's method, which is considered a reliable technique.

## Blood biochemical analysis

Blood samples underwent blood cell counts and biochemical analysis, including creatinine kinase (CK), aspartate aminotransferase, alanine aminotransferase, lactate dehydrogenase,

creatinine, blood urea nitrogen, hemoglobin A1c, total-cholesterol, low-density lipoprotein cholesterol, high-density lipoprotein cholesterol, and triglyceride. Blood biochemical analysis was performed by the clinical laboratory department in the Gifu Municipal Hospital or Gifu University Hospital.

### Drugs used and complications

Drugs used by the patients and complications were examined.

### Statistical analysis

Data are shown as the mean ± standard deviation. Categorical data were summarized as percentages and compared using chi-square or Fisher's exact test as appropriate. The significance of differences between 2 groups for variables that were normally distributed was determined by paired or unpaired Student's t-test. Significance of difference among 3 groups was determined by one way analysis of variance (ANOVA) followed by Dunnet method. Correlation coefficients between two variables were obtained by linear regression analysis using Pearson's correlation analysis. Statistical analyses were performed using GraphPad Prism 7 (GraphPad Software Inc.). A p-value $< 0.05$ was considered significant, and $p < 0.01$ and $p < 0.001$ were considered highly significant.

## Results

### Patients' baseline characteristics and drugs used in the control and AMI groups

Patients' baseline characteristics and drugs used are shown in Table 1.

The mean age was 69.2 ± 14.1 and 76.8 ± 8.5 years old in AMI group and Control group, respectively. There was no significant difference in age, sex, complications, biochemical data or drugs used between AMI group and Control group except for use of antiplatelet.

### Plasma progranulin levels

There was no significant difference in plasma progranulin levels between the control group (69.5 ± 24.6 ng/mL) and AMI group on day 0 (84.2 ± 47.1 ng/mL) after AMI (Fig 1A). However, plasma progranulin levels were significantly higher in the AMI group on day 7 (104.2 ± 52.0 ng/mL) than in the control group (69.5 ± 24.6 ng/mL) (p = 0.045) (Fig 2A). In patients with AMI in the acute phase, plasma progranulin levels significantly increased from 84.2 ± 47.1 ng/mL on day 0 to 104.2 ± 52.0 ng/mL on day 7 (p = 0.0002) (Fig 2B), and plasma progranulin levels on day 0 were closely and positively correlated with plasma progranulin levels on day 7 (r = 0.938, p < 0.0001) (Fig 2C).

### LVEF, LVSd, and LVDd in the acute and chronic phases after AMI

Cardiac echocardiography was performed in the acute (within 7 days) and chronic (6 months) phases of AMI. There were no significant differences between LVEF in the acute phase (52.5 ± 8.1%) and in the chronic phase (55.3 ± 10.9%) (Fig 3A) (meanΔLVEF ± SD was 2.806 ± 7.548%), between LVSd in the acute phase (32.6 ± 5.4 mm) and in the chronic phase (32.8 ± 6.3 mm) (Fig 3B), and between LVDd in the acute phase (45.3 ± 5.6 mm) and in the chronic phase (47.2 ± 6.3 mm) (Fig 3C).

**Table 1. Patients' baseline characteristics and drugs used.**

| | AMI group (n = 18) | Control group (n = 16) | P value |
|---|---|---|---|
| Age (years old) | 69.2±14.1 | 76.8±8.5 | 0.071 |
| Sex (n) | M/F (14/4) | M/F (10/6) | 0.457 |
| **Complications** | | | |
| HTN, n (%) | 14(77.8) | 14(87.5) | 0.660 |
| HL, n (%) | 4(22.2) | 7(43.8) | 0.274 |
| DM, n (%) | 6(33.3) | 4(25.0) | 0.715 |
| **Biochemical data** | | | |
| Creatinine (mg/dL) | 0.95±0.85 | 0.87±0.25 | 0.719 |
| TC (mg/dL) | 189.3±38.4 | 173.3±39.7 | 0.247 |
| LDL-C (mg/dL) | 123.5±38.6 | 99.8±27.8 | 0.059 |
| HDL-C (mg/dL) | 45.1±9.3 | 47.6±13.3 | 0.552 |
| TG (mg/dL) | 140.3±104.5 | 142.7±90.5 | 0.945 |
| HbA1C (%) | 6.4±0.9 | 6.0±0.7 | 0.107 |
| Hb (g/dL) | 13.7±2.1 | 13.2±1.7 | 0.416 |
| **Drugs used,** n (%) | | | |
| ACEI/ARB | 14(78) | 11(69) | 0.703 |
| CCB | 4(22) | 11(69) | 0.274 |
| BB | 8(44) | 4 (44) | >0.999 |
| Statin | 15(83) | 9(56) | 0.134 |
| DPP4-I | 1(6) | 2(13) | 0.591 |
| Metformin | 2(11) | 1(6) | >0.999 |
| Antiplatelet | 18(100): | 7(44) | 0.0002 |

HTN = hypertension, HL = hyperlipidemia, DM = diabetes mellitus, TC = total cholesterol, LDL-C = low density lipoprotein cholesterol, HDL-C = high density lipoprotein cholesterol, TG = triglyceride, HbA1c = hemoglobin A1c, Hb = hemoglobin, ACEI = angiotensin- converting enzyme inhibitor, ARB = angiotensin II receptor blocker, CCB = calcium channel blocker, BB = beta blocker, DPP4-I = dipeptidyl peptidase 4 inhibitor.

## Relationship between the increase in plasma progranulin levels in the acute phase and the increase in LVEF, LVSd, and LVDd in the chronic phase of 6 months

The increase of plasma progranulin levels in the acute phase (ρprogranulin: plasma progranulin level on day 7 –plasma progranulin level on day 0) was positively correlated with the increase in LVEF between the acute and chronic phases (ρEF: LVEF in the chronic phase at 6 months–LVEF in the acute phase) (Y = 1.135X + 16.97, r = 0.4725, p = 0.0238) (Fig 4A). There was no significant correlation between ρprogranulin and changes in LVSd between the acute and chronic phases (ρLVSd: LVSd in the chronic phase at 6 months–LVSd in the acute phase) (Y = -1.311X + 20.52, r = 0.2826, p = 0.2558) (Fig 4B). There was no significant correlation between ρprogranulin and the changes in LVDd between the acute and chronic phases (ρLVDd: LVDd in the chronic phase at 6 months–LVDd in the acute phase) (Y = -1.274X + 22.63, r = 0.2947, p = 0.2352) (Fig 4C).

## The increase in plasma progranulin levels in the acute phase and the increase in LVEF (ρLVEF<0 or ρLVEF≥0) at 6 months

Out of 18 AMI patients, ten showed an increase and eight showed a decrease in LVEF in the chronic phase at 6 months. Subsequently, we divided the patients into two groups: ρLVEF<0

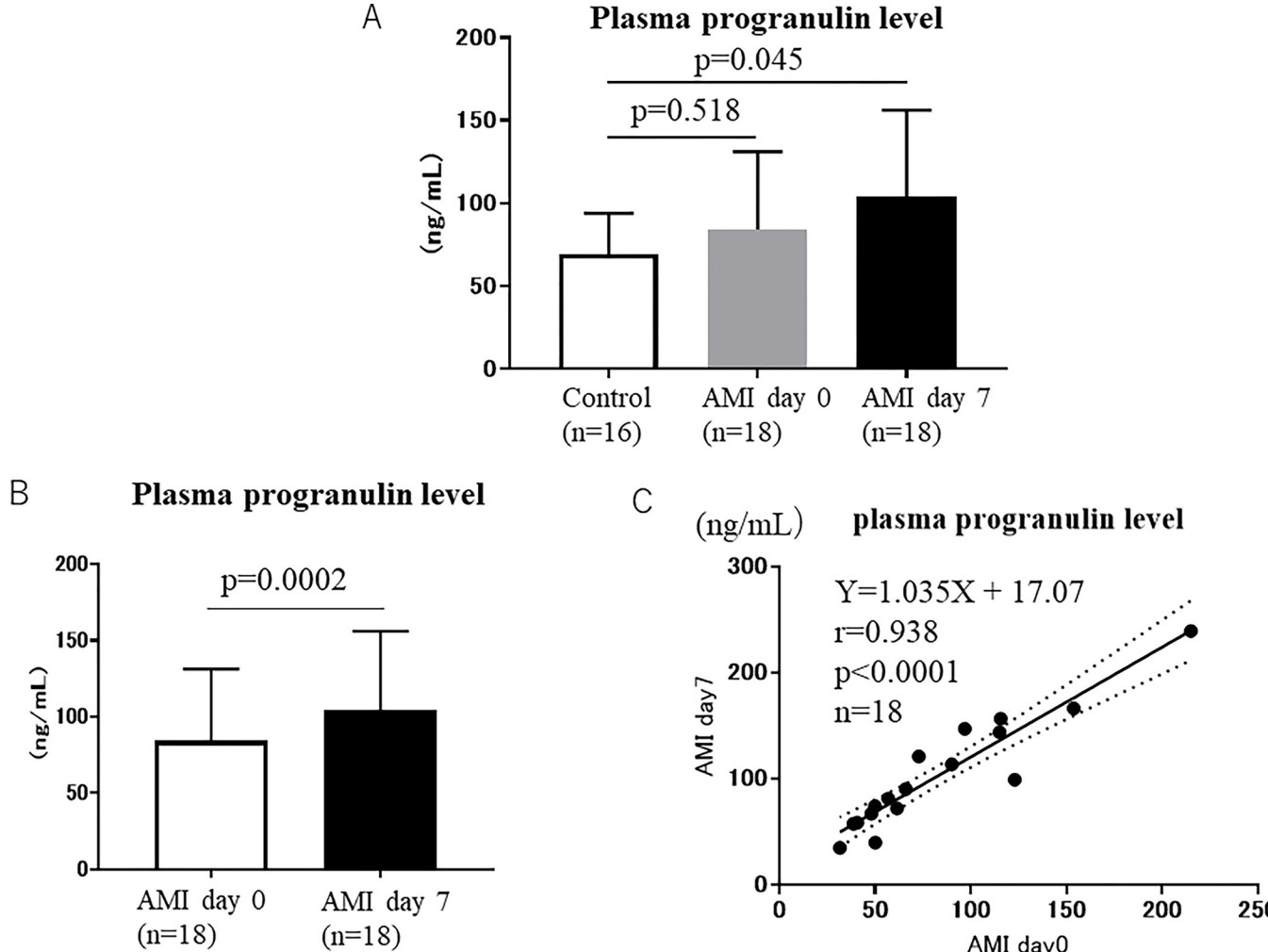

**Fig 2.** A: Plasma progranulin levels in the control and on day 0 in the AMI group. B: Plasma progranulin levels on day 0 and day 7 in the AMI group. C: Relationship of plasma progranulin levels between days 0 and 7 in the AMI group.

group in which LVEF decreased in the chronic phase at 6 months, and ρLVEF≥0 group in which LVEF increased or remained unchanged in the chronic phase at 6 months. The increase of plasma progranulin levels (ρprogranulin) in the acute phase was significantly higher in the ρLVEF≥0 group (30.35 ± 3.68 ng/mL) than in the ρLVEF<0 group (7.41 ± 6.0 ng/mL) (p = 0.0037) (Fig 5).

## Factors that may affect LVEF in the chronic phase at 6 months

Factors that may affect LVEF were compared between the ΔLVEF-negative and -positive groups at 6 months after AMI (Table 2). Among these factors, only Δprogranulin was significantly higher in the ΔLVEF-positive (ΔLVEF≥0) group than the ΔLVEF-negative (ΔLVEF≺0) group (p = 0.0037). There were no differences in age; sex; presence of hypertension, hyperlipidemia, or diabetes mellitus; use of drugs, such as ARB/ACEI, beta-blockers, Ca-blockers, diuretics, nitrates, statins, EPA, aspirin, clopidogrel, metformin, or DPP4-inhibitors; alcohol; smoking; or peak CK (Table 2). In the univariate analysis, only Δprogranulin was significantly correlated with LV functional recovery in the chronic phase at 6 months after AMI.

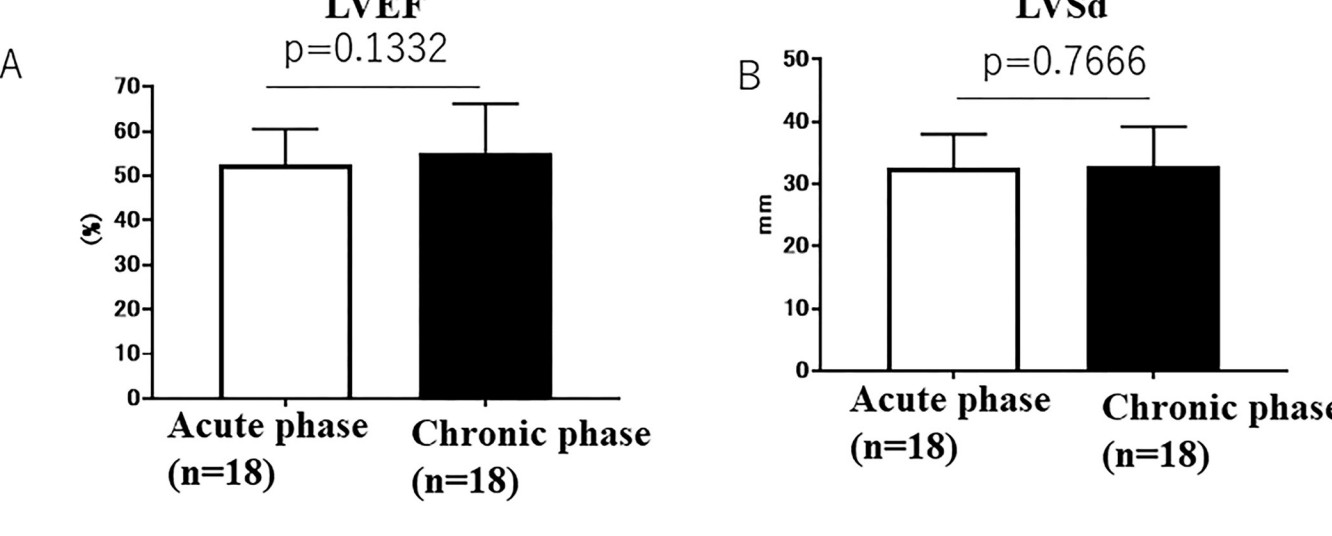

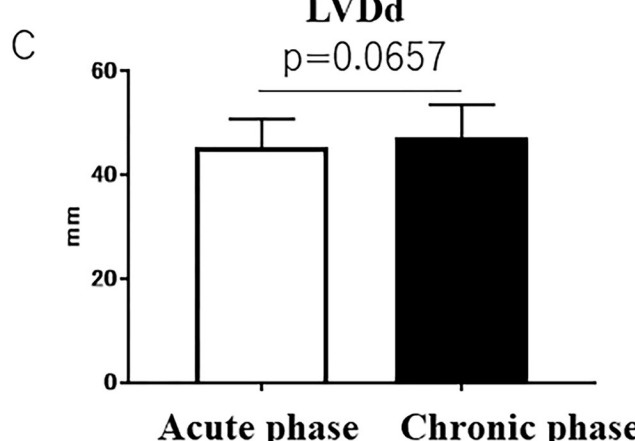

**Fig 3.** A: LVEF in the acute and chronic phases. B: LVSd in the acute and chronic phases. C: LVDd in the acute and chronic phases.

## Discussion

In the present study, we found for the first time that plasma progranulin levels increase in the acute phase on day 7 after AMI and that the increase in plasma progranulin levels ($\rho$PGN) in the acute phase is positively correlated with the increase in LVEF ($\rho$EF) between the chronic and acute phases.

A previous study reported that progranulin deficiency exacerbated tissue injury in a murine model of renal ischemia-reperfusion injury [12]. Moreover, the expression of progranulin significantly increased in the myocardial ischemic area in a murine model of AMI [10]. AMI leads to the infiltration of leukocytes, including macrophages, into the myocardial infarct border areas, promoting phagocytosis to remove necrotic cardiomyocytes and matrix debris [13,14]. Progranulin is mainly secreted from macrophages and neutrophils and attenuates inflammation of the tissue injury [15]; thus, infiltrated neutrophils and macrophages in the infarct border area may have secreted progranulin, suppressed the inflammation, and repaired the infarcted myocardium in patients with AMI.

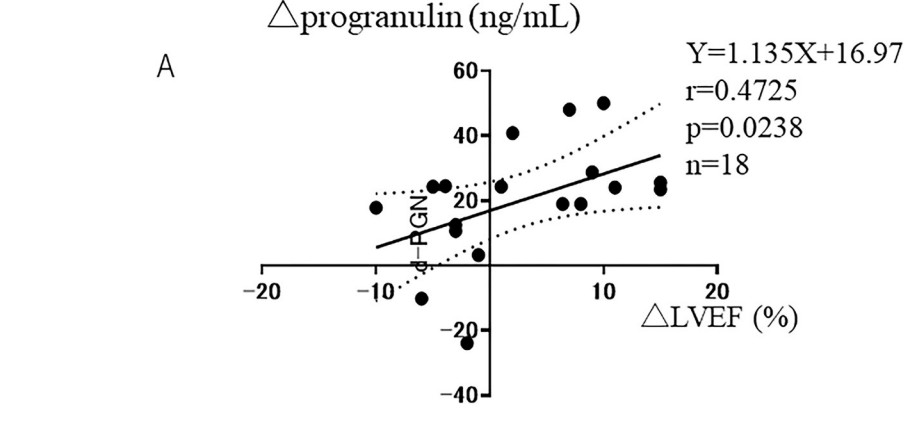

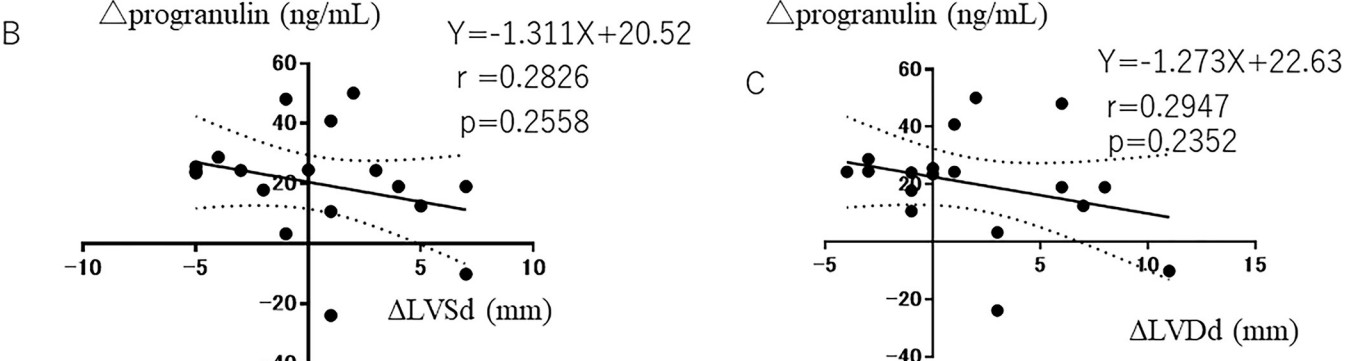

**Fig 4. A:** Relationship between Δplasma progranulin levels and ΔLVEF between acute and chronic phases. **B:** Relationship between Δplasma progranulin levels and ΔLVSd between acute and chronic phases. **C:** Relationship between Δplasma progranulin levels and ΔLVDd between acute and chronic phases.

A previous study reported that plasma progranulin levels were elevated in patients with AMI compared with healthy controls [16]; however, another study reported that there was no difference in plasma progranulin levels between patients with acute coronary syndrome and controls [17]. In the present study, there was no significant difference in plasma progranulin levels between AMI patients on day 0 and the control; however, plasma progranulin levels were significantly higher in the AMI group on day 7 compared with the control group (Fig 1A).

In patients with AMI, plasma progranulin levels significantly increased from $84.2 \pm 47.1$ ng/mL on day 0 to $104.2 \pm 52.0$ ng/mL on day 7 ($p = 0.0002$) after admission to the hospital (Fig 1B). Moreover, plasma progranulin levels on day 0 were closely and positively correlated with plasma progranulin levels on day 7 ($r = 0.9439$, $Y = 1.025X + 18.03$, $p < 0.0001$) (Fig 1C), suggesting that plasma progranulin levels increase in a constant ratio from day 0 to day 7 in the acute phase of AMI. According to a previous animal study [10], the expression of progranulin in the ischemic area significantly increases on days 1, 3 and 5 after AMI. This suggests that it takes time until plasma progranulin levels increase after AMI.

The mechanism by which plasma progranulin levels increase within 7 days after AMI may be explained by the infiltration of neutrophils and macrophages into the infarct border area within 7 days in the acute phase of AMI [10,14] and secretion of progranulin, which then overflows into the peripheral blood within 7 days after AMI. Therefore, the increase in plasma progranulin levels may have reflected the secretion of progranulin from the neutrophils and macrophage infiltration into the infarct border area because of myocardial tissue damage due to AMI.

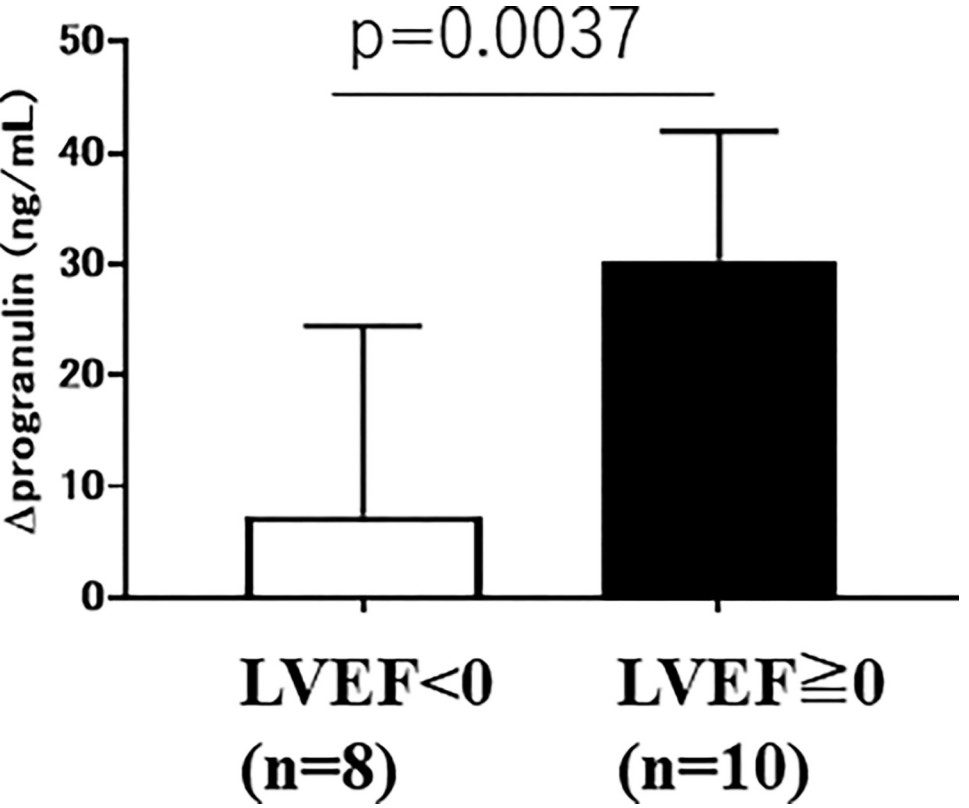

**Fig 5. The increase in plasma progranulin levels in the acute phase and the increase in LVEF (ρLVEF<0 or ρLVEF≥0) at 6 months.**

We examined whether the increase in plasma progranulin levels in the acute phase affected cardiac function in the chronic phase 6 months after the onset of AMI. We measured LVEF in the acute phase and 6 months after the onset of AMI because the long-term clinical outcome of AMI after PCI is determined by the recovery of LVEF. A previous study reported that good recovery of LVEF was associated with a good long-term prognosis, whereas poor recovery was associated with a poor long-term prognosis [17,18]. We investigated the relationship between the increase in plasma progranulin levels in the acute phase (plasma progranulin level on day 7 –plasma progranulin level on day 0) and increases in LVEF, LVSd, and LVDd between acute and chronic phases (ρEF, ρLVSd and ρLVDd: 6 months of chronic phase–acute phase). We found the increase in plasma progranulin levels in the acute phase (ρprogranulin: difference between day 0 and day 7) was positively correlated with the increase in LVEF between the acute and chronic phases (Y = 1.135X + 16.97, r = 0.4725, p = 0.0238), suggesting that a greater increase in plasma progranulin levels in the acute phase is associated with a greater improvement of LVEF in the chronic phase after the onset of AMI (Fig 3). This suggests that the increase in plasma progranulin levels is associated with the improvement of cardiac function, as assessed by LVEF, in patients with AMI. This supports our previous animal experimental data that found that intravenous administration of recombinant human progranulin improves LV function in murine and rabbit models of AMI [10]. The precise mechanism by which progranulin improved LV function in the present clinical study remains unclear. However, previous animal study demonstrated that the administration of recombinant progranulin significantly attenuated the infiltration of neutrophils and cardiac fibrosis in the infarct border

**Table 2. Factors affecting △LVEF between acute and chronic phases.**

| Characteristics | Overall, N = 18 | LVEF<0, N = 8 | LVEF≧0, N = 10 | P value |
|---|---|---|---|---|
| Age | 69.2±14.1 | 69.1±16.1 | 69.2±13.2 | 0.992 |
| Sex | female 4(22%) | 2(25%) | 2(20%) | >0.999 |
| | male 14(78%) | 6(75%) | 8 (80%) | |
| Day 0-progranulin | 80.9±50.4 | 86.5±70 | 76.3±30.4 | 0.681 |
| Day 7-progranulin | 100.95±4.7 | 93.9±73.0 | 106.4±37.3 | 0.644 |
| △progranulin | 20.2±18.1 | 7.4±17.0 | 30.4±11.6 | 0.0037 |
| Peak CK | 2,482±1,886 | 2,538±2,612 | 2,437±1,188 | 0.914 |
| Hypertension | Absent 4(22%) | 3(38%) | 1(10%) | 0.275 |
| | Present 14(78%) | 5(62%) | 9(90%) | |
| Hyperlipidemia | Absent 3(17%) | 1(12%) | 2(20%) | >0.999 |
| | Present 15(83%) | 7(88%) | 8(80%) | |
| Diabetes Mellitus | Absent 12(67%) | 6(60%) | 4(50%) | >0.999 |
| | Present 6(33%) | 4(40%) | 4(50%) | |
| ACEI/ARB | Absent 4(22%) | 2(22%) | 2(22%) | >0.999 |
| | Present 14(78%) | 7(78%) | 7(78%) | |
| Nitrate | Absent 17(94%) | 8(100%) | 9(90%) | >0.999 |
| | Present 1(6%) | 0(0%) | 1(10%) | |
| CCB | Absent 14(78%) | 7(88%) | 7(70%) | 0.588 |
| | Present 4(22%) | 1(12%) | 3(30%) | |
| Beta blocker | Absent 10(56%) | 3(38%) | 7(70%) | 0.342 |
| | Present 8(44%) | 5(62%) | 3(30%) | |
| Diuretics | Absent 15(83%) | 7(88%) | 7(70%) | 0.588 |
| | Present 3(17%) | 1(12%) | 3(30%) | |
| Statins | Absent 3(17%) | 1(12%) | 2(20%) | >0.999 |
| | Present 15(83%) | 7(88%) | 8(80%) | |
| Aspirin | Absent 3(17%) | 1(12%) | 2(20%) | >0.999 |
| | Present 15(83%) | 7(88%) | 8(80%) | |
| Clopidogrel | Absent 1(16%) | 1(12%) | 0(0%) | 0.444 |
| | Present 17(84%) | 7(88%) | 10(100%) | |
| Metformin | Absent 16(89%) | 6(75%) | 10(100%) | 0.183 |
| | Present 2(11%) | 2(25%) | 0(0%) | |
| DPP4 inhibitors | Absent 17(94%) | 7(88%) | 10(100%) | 0.444 |
| | Present 1(6%) | 1(12%) | 0(0%) | |
| Smoking | Absent 5(32%) | 1(12%) | 4(40%) | 0.314 |
| | Present 13(68%) | 7(88%) | 6(60%) | |
| Alcohol | Absent 14(78%) | 7(88%) | 7(70%) | 0.588 |
| | Present 4(22%) | 1(12%) | 3(30%) | |

CCB = calcium channel blocker, DPP4 = dipeptidyl peptidase 4, ARB = angiotensin 2 receptor antagonist, ACEI = angiotensin converting enzyme inhibitor.

areas, and then attenuated the infarct size and ameliorated the cardiac dysfunction [10]. Therefore, it may be possible that attenuation of neutrophil infiltration and fibrosis in the infarct border area improved the cardiac function at the chronic phase of 6 months after AMI in the present study.

Concerning LV dilation, the increase in plasma progranulin levels in the acute phase had no significant effect on changes in LVSd or LVDd, although a negative correlation between the increase in plasma progranulin levels and LVSd or LVDd was observed (Fig 4). A previous study reported that infarct size is associated with LV dilatation [19]. Thus, the reason plasma

progranulin levels did not affect LVSd or LVDd in the chronic phase (Fig 4A and 4B) may be that AMI was not severe enough to cause LV remodeling in the chronic phase, given the initial mean LVEF of 52.5 ± 8.1%.

As shown in Table 2, among many factors that may affect the recovery of LV function, univariate analysis demonstrated that only ρprogranulin was correlated with improvement of LV function in the chronic phase at 6 months. This suggests that the increase in plasma progranulin levels in the acute phase was associated with the improvement of LVEF in the chronic phase in the AMI patients.

The increase in progranulin levels at day 7 is statistically significant but quite small. This is presumably because progranulin release is very local at the margins of the ischemic injury and is diluted in samples taken from the general circulation. However, study limitation of the present study was that the number of AMI patients was small. A clinical study with a larger number of AMI patients is warranted.

In conclusion, we demonstrated that plasma progranulin levels increased in the acute phase after AMI, and that the increase in plasma progranulin levels was positively correlated with the increase in LVEF in the chronic phase. The increase in plasma progranulin levels may contribute to improvement of LV function in the chronic phase in patients with AMI. The increase in plasma progranulin levels in the acute phase may serve as a biomarker to predict the recovery of LV function in the chronic phase in patients with AMI.

## Supporting information

**S1 Checklist. CONSORT 2010 checklist of information to include when reporting a randomised trial\*.**
(DOC)

**S1 File.**
(DOCX)

## Acknowledgments

We thank Mrs. Kaori Osawa (Gifu Municipal Hospital, Gifu, Japan) and Miss Akiko Tsujimoto for their technical support.

## Author Contributions

**Conceptualization:** Shinya Minatoguchi.

**Data curation:** Shingo Minatoguchi, Atsushi Satake, Hirotaka Murase, Ryo Yoshizumi, Hisaaki Komaki, Shinya Baba, Shinji Yasuda, Shinsuke Ojio, Toshiki Tanaka, Shinya Minatoguchi.

**Funding acquisition:** Shinya Minatoguchi.

**Investigation:** Shingo Minatoguchi, Atsushi Satake, Hirotaka Murase, Shinya Minatoguchi.

**Methodology:** Shingo Minatoguchi, Shinya Minatoguchi.

**Supervision:** Hiroyuki Okura, Shinya Minatoguchi.

**Validation:** Shingo Minatoguchi, Shinya Minatoguchi.

**Writing – original draft:** Shingo Minatoguchi.

**Writing – review & editing:** Shinya Minatoguchi.

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
