## [Decision Letter · Decision Letter 0]

29 Aug 2024

PONE-D-24-28401Elevated plasma progranulin levels in the acute phase are correlated with recovery of left ventricular function in the chronic phase in patients with acute myocardial infarctionPLOS ONE

Dear Dr. Minatoguchi,

Thank you for submitting your manuscript to PLOS ONE. After careful consideration, we feel that it has merit but does not fully meet PLOS ONE’s publication criteria as it currently stands. Therefore, we invite you to submit a revised version of the manuscript that addresses the points raised during the review process.

Please submit your revised manuscript by Oct 13 2024 11:59PM. If you will need more time than this to complete your revisions, please reply to this message or contact the journal office at plosone@plos.org. Please include the following items when submitting your revised manuscript:A rebuttal letter that responds to each point raised by the academic editor and reviewer(s). You should upload this letter as a separate file labeled 'Response to Reviewers'.A marked-up copy of your manuscript that highlights changes made to the original version. You should upload this as a separate file labeled 'Revised Manuscript with Track Changes'.An unmarked version of your revised paper without tracked changes. You should upload this as a separate file labeled 'Manuscript'.If applicable, we recommend that you deposit your laboratory protocols in protocols.io to enhance the reproducibility of your results. Protocols.io assigns your protocol its own identifier (DOI) so that it can be cited independently in the future. For instructions see: https://journals.plos.org/plosone/s/submission-guidelines#loc-laboratory-protocols. Additionally, PLOS ONE offers an option for publishing peer-reviewed Lab Protocol articles, which describe protocols hosted on protocols.io. Read more information on sharing protocols at https://plos.org/protocols?utm_medium=editorial-email&utm_source=authorletters&utm_campaign=protocols.

We look forward to receiving your revised manuscript.

Kind regards,

Wannaporn Ittiprasert, Ph.D

Academic Editor

PLOS ONE

Journal Requirements:

Reviewers' comments:

Reviewer's Responses to Questions

**Comments to the Author**

1. Is the manuscript technically sound, and do the data support the conclusions?

Reviewer #1: Yes

Reviewer #2: Yes

Reviewer #3: Yes

Reviewer #4: Partly

2. Has the statistical analysis been performed appropriately and rigorously? 

Reviewer #1: Yes

Reviewer #2: Yes

Reviewer #3: Yes

Reviewer #4: I Don't Know

3. Have the authors made all data underlying the findings in their manuscript fully available?

Reviewer #1: Yes

Reviewer #2: Yes

Reviewer #3: Yes

Reviewer #4: Yes

4. Is the manuscript presented in an intelligible fashion and written in standard English?

Reviewer #1: Yes

Reviewer #2: Yes

Reviewer #3: Yes

Reviewer #4: Yes

5. Review Comments to the Author

Reviewer #1: The topic is very interesting, it will enhance the basic knowledge about the progranulin role in AMI.

‘The mean age of AMI patients was 69.2 ± 14.1 years old’, and ‘The mean age of the control was 76.8 ± 8.5 years old’; should be explained in the results.

Improve the spelling of ELISA, authors have written ‘ELIZA’.

It is suggested to briefly explain how Progranulin causes pathophysiological aspects in AMI.

First, explain the full form of terms then use its abbreviation.

English should be improved.

Blood biochemical analysis (How did authors measure this biochemical analysis; did they use ELISA, or which instrument did they use, while performing these parameters?)

Statistical analysis

1. What tests, the authors used to normalize the data

2. When correlation analysis was applied, what were the two variables?

Reviewer #2: Interesting paper. Some issues should be added

-abstract: numeric data should be added about levels of granulin

- Complete revascularization has shown to reduce risk of HF after ACS: this should be add and commented (quote on PMID: 37489724)

- mean of delta LVEF should be added to understant if clinically significant

Reviewer #3: Progranulin, a secreted glycoprotein that is often associated with tissue regeneration and remodeling, has been implicated in recovery after ischemic events. The authors of the current paper have published interesting research that strongly supports that progranulin improves some aspects of recovery after myocardial ischemia in animal models, however there is very little information available to assess the role of progranulin in recovery after acute myocardial ischemia (AMI) in patients. The authors address this issue by measuring circulating progranulin levels on day zero and day seven after admission of patients experiencing clinically confirmed AMI. These values are then correlated with metrics of cardia function six months later.

The authors find an increase in circulating progranulin on day seven of AMI but not on day zero. The control group was patients experiencing chest pains for reasons other than AMI. The baseline characteristics of the AMI and control group were comparable except for use of “antiplatelet” medication which was highly statistically different between the AMI and controls. Briefly, taking the group as a whole, there was no change in left ventricular ejection fraction (LVEF), LV end-systolic dimension (LVSd) and LV end-diastolic dimension (LVDd) among the patient group between the acute and chronic (6 month) stages. However, when the patients were analyzed in terms of delta PGRN (the difference between progranulin control and day seven value), low delta PGRN values showed no improvement in LVEF while patients with high delta PGRN displayed improvements in LVEF at six months. Based on their animal work the authors suggest the improvement might result from recruitment of neutrophils to the margins on the ischemic tissue and local release of progranulin. It was noteworthy that of 20 variables that might have affected recovery of LVEF only delta PGRN achieved statistical significance.

Comments.

There are reports of progranulin levels increasing in ischemia, but this work is novel because it shows a correlation between higher progranulin and improved functional tissue recovery in patients. The increase in progranulin levels at day seven is statistically significant but quite small. This is presumably because progranulin release is very local (at the margins of the ischemic injury) and is diluted in samples taken from the general circulation.

1. Patients were divided into delta LVEF <0 (LVEF decreased in the chronic phase) and delta LVEF≥0 (LVEF increased or stayed the same). It is difficult however to judge what this really means; in the delta LVEF≥0 how many LVEFs were unchanged; what was the distribution among those that increased (did one or two show big changes and others small changes, were all changes modest, was there a single outlier that might distort the data etc.). Furthermore, did the size of the individual delta LVEF≥0 values correlate well with the corresponding delta PGRN (did larger delta LVEF≥0 values also have larger delta PGRNs)?

2. The sample size of the study is quite small (eighteen patients). This should be pointed out in the discussion as a potential limitation.

Reviewer #4: Comments

1) The acronym for enzyme-linked immunosorbent assay is misspelled as ELIZA. Replace with ELISA in manuscript. Also, the word must be spelled out in its first use.

2) Describe the results of Table 1. The first paragraph in the results section has no information in it except to look at Table 1.

3) The data do not support the title that elevated progranulin correlates with recovery. The standard deviation error bars significantly overlap, putting into question the significance of the findings. The number of patients should be increased to improve confidence in the findings.

4) Please confirm with a statistician the correct statistical tests were carried out with the appropriate conclusions.

6. PLOS authors have the option to publish the peer review history of their article (what does this mean?). If published, this will include your full peer review and any attached files.

Reviewer #1: **Yes: **DR Saira Rafaqat

Reviewer #2: **Yes: **Fabrizio D'Ascenzo

Reviewer #3: No

Reviewer #4: No

---

## [Author Response · Author response to Decision Letter 0]

6 Sep 2024

Review Comments to the Author

Reviewer #1: The topic is very interesting, it will enhance the basic knowledge about the progranulin role in AMI.

‘The mean age of AMI patients was 69.2 ± 14.1 years old’, and ‘The mean age of the control was 76.8 ± 8.5 years old’; should be explained in the results.

As suggested by the reviewer, wｅ explained the mean age ±SD of AMI and control groups in the Results section as follows on page 7, paragraph 1, lines 3-4 : 

“The mean age was 69.2 ± 14.1 and 76.8 ± 8.5 years old in AMI group and Control group, respectively.”

Improve the spelling of ELISA, authors have written ‘ELIZA’.

As suggested by the reviewer, the spelling of ELIZA was corrected to ELISA in the Method section and abstract.

It is suggested to briefly explain how Progranulin causes pathophysiological aspects in AMI.

As suggested by the reviewer, we discussed the mechanism by which progranulin improved the LV function after AMI in the Discussion section as follows on page 12, paragraph 1, lines 10-17.

“The precise mechanism by which progranulin improved LV function in the present clinical study remains unclear. However, previous animal study demonstrated that the administration of recombinant progranulin significantly attenuated the infiltration of neutrophils and cardiac fibrosis in the infarct border areas, and then attenuated the infarct size and ameliorated the cardiac dysfunction (10). Therefore, it may be possible that attenuation of neutrophil infiltration and fibrosis in the infarct border area improved the cardiac function at the chronic phase of 6 months after AMI in the present study.” 

First, explain the full form of terms then use its abbreviation.

As suggested by the reviewer, we first explained the full form of terms and then used its abbreviation in the manuscript. 

English should be improved.

English has been checked by the native English speaker.

Blood biochemical analysis (How did authors measure this biochemical analysis; did they use ELISA, or which instrument did they use, while performing these parameters?)

Thank you for the comment. Blood biochemical analysis was performed by the clinical laboratory department in the Gifu Municipal Hospital or Gifu University Hospital. This was added to the Method section as follows on page 6, paragraph 3, lines 5-6.

“Blood biochemical analysis was performed by the clinical laboratory department in the Gifu Municipal Hospital or Gifu University Hospital.” 

Statistical analysis

1. What tests, the authors used to normalize the data.

We stated the name of the test to perform the statistical analysis in details in the Method section as follows on page 6, paragraph 5, on page 7, paragraph 1.

“Statistical analysis Data are shown as the mean ± standard deviation. Categorical data were summarized as percentages and compared using chi-square or Fisher’s exact test as appropriate. The significance of differences between 2 groups for variables that were normally distributed was determined by paired or unpaired Student’s t-test. Significance of difference among 3 groups was determined by one way analysis of variance (ANOVA) followed by Dunnet method. Correlation coefficients between two variables were obtained by linear regression analysis using Pearson’s correlation analysis. Statistical analyses were performed using GraphPad Prism 7 (GraphPad Software Inc.). A p-value < 0.05 was considered significant, and p < 0.01 and p < 0.001 were considered highly significant.”

2. When correlation analysis was applied, what were the two variables?

The analysis of the data were performed after the completion of the data collection (AMI group: n=18, control group: n=16). 

As shown in Fig. 4,A, B and C, the two variables when correlaton analysis was applied were between Δprogranurin and ΔLVEF, between Δprogranulin and ΔLVSd, and between Δprogranulin and ΔLVDd. 

Reviewer #2: Interesting paper. Some issues should be added

-abstract: numeric data should be added about levels of granulin

As suggested by the reviewer, numeric data on levels of progranulin were added in the abstract as follows:

 “Plasma progranulin levels were measured by enzyme-linked immunosorbent assay. Echocardiography was performed in the acute (within 7 days) and chronic (6 months) phases of AMI to evaluate left ventricular ejection fraction using the modified Simpson’s method. Plasma progranulin levels in the AMI group on day 0 (69.5 ± 24.6 ng/mL) were similar to those in the control group (84.2 ± 47.1 ng/mL). There was a significant increase in progranulin levels in the AMI group on day 7 (104.2 ± 52.0 ng/mL) compared with day 0 (84.2 ± 47.1 ng/mL).”

- Complete revascularization has shown to reduce risk of HF after ACS: this should be add and commented (quote on PMID: 37489724)

As suggested by the reviewer, we added the sentence that complete revascularization of the occluded coronary artery was performed in all the AMI patients in the AMI group and commented this sentence in the Method section and quoted the PMID:37489724 as a reference 11. 

In the Method section: on page 5, paragraph 1, lines 3-6.

“All the AMI patients were performed complete revascularization of the occluded coronary artery, which has been reported to reduce the risk of heart failure hospitalization and cardiovascular death (11).”

- mean of delta LVEF should be added to understant if clinically significant

As suggested by the reviewer, we added the mean of delta LVEF in the AMI group (n=18) in the Result section as follows: On page 8, paragraph 2, lines 3-4.

“(meanΔLVEF was 2.806 ± 7.548 %)”

Reviewer #3: Progranulin, a secreted glycoprotein that is often associated with tissue regeneration and remodeling, has been implicated in recovery after ischemic events. The authors of the current paper have published interesting research that strongly supports that progranulin improves some aspects of recovery after myocardial ischemia in animal models, however there is very little information available to assess the role of progranulin in recovery after acute myocardial ischemia (AMI) in patients. The authors address this issue by measuring circulating progranulin levels on day zero and day seven after admission of patients experiencing clinically confirmed AMI. These values are then correlated with metrics of cardia function six months later.

The authors find an increase in circulating progranulin on day seven of AMI but not on day zero. The control group was patients experiencing chest pains for reasons other than AMI. The baseline characteristics of the AMI and control group were comparable except for use of “antiplatelet” medication which was highly statistically different between the AMI and controls. Briefly, taking the group as a whole, there was no change in left ventricular ejection fraction (LVEF), LV end-systolic dimension (LVSd) and LV end-diastolic dimension (LVDd) among the patient group between the acute and chronic (6 month) stages. However, when the patients were analyzed in terms of delta PGRN (the difference between progranulin control and day seven value), low delta PGRN values showed no improvement in LVEF while patients with high delta PGRN displayed improvements in LVEF at six months. Based on their animal work the authors suggest the improvement might result from recruitment of neutrophils to the margins on the ischemic tissue and local release of progranulin. It was noteworthy that of 20 variables that might have affected recovery of LVEF only delta PGRN achieved statistical significance.

Comments.

There are reports of progranulin levels increasing in ischemia, but this work is novel because it shows a correlation between higher progranulin and improved functional tissue recovery in patients. The increase in progranulin levels at day seven is statistically significant but quite small. This is presumably because progranulin release is very local (at the margins of the ischemic injury) and is diluted in samples taken from the general circulation.

1. Patients were divided into delta LVEF <0 (LVEF decreased in the chronic phase) and delta LVEF≥0 (LVEF increased or stayed the same). It is difficult however to judge what this really means; in the delta LVEF≥0 how many LVEFs were unchanged; 

In the delta LVEF≥0 group (n=10), LVEF increased in all 10 patients, and in the delta LVEF<0 group (n=8), LVEF decreased in all 8 patients. There was no patient with LVEF unchanged at 6 months after the onset of AMI. 

what was the distribution among those that increased (did one or two show big changes and others small changes, were all changes modest, was there a single outlier that might distort the data etc.). 

The distribution of the delta LVEF in the LVEF≥0 group was 15, 15, 11, 10, 9, 8, 7, 6.4, 2, 1(mean±SD = 8.44±1.485).

The distribution of the delta LVEF in the LVEF<0 group was -10, -6, -5, -3.9, -3, -3, -2, -1 (mean±SD = -4.238±2.817).

Furthermore, did the size of the individual delta LVEF≥0 values correlate well with the corresponding delta PGRN (did larger delta LVEF≥0 values also have larger delta PGRNs)?

The answer of this question is presented in Fig. 4-A. As shown in Fig. 4-A, larger delta PGRNs positively correlated with larger delta LVEF (Y=1.135X + 16.97, r=0.4725, p=0.0238, n=18). This suggests that ΔLVEF correlated well with the corresponding ΔPGRN. 

2. The sample size of the study is quite small (eighteen patients). This should be pointed out in the discussion as a potential limitation.

As suggested by the reviewer, we pointed out that the sample size is quite small in the study limitation in the Discussion section as follows: on page 13, paragraph 2, lines 1-5.

“The increase in progranulin levels at day 7 is statistically significant but quite small. This is presumably because progranulin release is very local at the margins of the ischemic injury and is diluted in samples taken from the general circulation. However, study limitation of the present study was that the number of AMI patients was small. A clinical study with a larger number of AMI patients is warranted.” 

Reviewer #4: Comments

1) The acronym for enzyme-linked immunosorbent assay is misspelled as ELIZA. Replace with ELISA in manuscript. Also, the word must be spelled out in its first use.

As suggested by the reviewer，ELIZA was replaced with ELISA. Enzyme-linked immunosorbent assay was spelled out in the first use and then ELISA in the text. 

2) Describe the results of Table 1. The first paragraph in the results section has no information in it except to look at Table 1.

As suggested by the reviewer, we described the results of Table 1 as follows: on page 7, paragraph 2, lines 2-5. 

“The mean age was 69.2 ± 14.1 and 76.8 ± 8.5 years old in AMI group and Control group, respectively. There was no significant difference in age, sex, complications, biochemical data or drugs used between AMI group and Control group except for use of antiplatelet.”

3) The data do not support the title that elevated progranulin correlates with recovery. The standard deviation error bars significantly overlap, putting into question the significance of the findings. The number of patients should be increased to improve confidence in the findings.

As suggested by the reviewer, the number of AMI patients is small. Therefore, we described this in the study limitation in the Method section as follows: On page 13, paragraph 2, lines 1-5.

“The increase in progranulin levels at day 7 is statistically significant but quite small. This is presumably because progranulin release is very local at the margins of the ischemic injury and is diluted in samples taken from the general circulation. However, study limitation of the present study was that the number of AMI patients was small. A clinical study with a larger number of AMI patients is warranted.” 

4) Please confirm with a statistician the correct statistical tests were carried out with the appropriate conclusions.

Thank you very much for the comment. We confirmed with a statistician that our statistical test is correct, and we added the statement more precisely in the Statistical analysis in the Method section as follows: on page 7, paragraph1, lines 2-5.

“The significance of differences between 2 groups for variables that were normally distributed was determined by paired or unpaired Student’s t-test. Significance of difference among 3 groups was determined by one way analysis of variance (ANOVA) followed by Dunnet method.”

---

## [Decision Letter · Decision Letter 1]

17 Oct 2024

Elevated plasma progranulin levels in the acute phase are correlated with recovery of left ventricular function in the chronic phase in patients with acute myocardial infarction

PONE-D-24-28401R1

Dear Dr. Shinya Minatoguchi,

We’re pleased to inform you that your manuscript has been judged scientifically suitable for publication and will be formally accepted for publication once it meets all outstanding technical requirements.

Kind regards,

Wannaporn Ittiprasert, Ph.D

Academic Editor

PLOS ONE

Additional Editor Comments (optional):

Reviewers' comments:

Reviewer's Responses to Questions

**Comments to the Author**

1. If the authors have adequately addressed your comments raised in a previous round of review and you feel that this manuscript is now acceptable for publication, you may indicate that here to bypass the “Comments to the Author” section, enter your conflict of interest statement in the “Confidential to Editor” section, and submit your "Accept" recommendation.

Reviewer #1: All comments have been addressed

Reviewer #2: All comments have been addressed

Reviewer #3: All comments have been addressed

2. Is the manuscript technically sound, and do the data support the conclusions?

Reviewer #1: Yes

Reviewer #2: (No Response)

Reviewer #3: Yes

3. Has the statistical analysis been performed appropriately and rigorously? 

Reviewer #1: Yes

Reviewer #2: (No Response)

Reviewer #3: I Don't Know

4. Have the authors made all data underlying the findings in their manuscript fully available?

Reviewer #1: Yes

Reviewer #2: (No Response)

Reviewer #3: Yes

5. Is the manuscript presented in an intelligible fashion and written in standard English?

Reviewer #1: Yes

Reviewer #2: (No Response)

Reviewer #3: Yes

6. Review Comments to the Author

Reviewer #1: Thank you again for your hard work. Authors have well explained all the comments in the revise manuscript.

Reviewer #2: (No Response)

Reviewer #3: The authors have answered the questions raised in the initial review and made appropriate changes in the text.

7. PLOS authors have the option to publish the peer review history of their article (what does this mean?). If published, this will include your full peer review and any attached files.

Reviewer #1: **Yes: **Dr. Saira Rafaqat

Reviewer #2: **Yes: **Fabrizio D'Ascenzo

Reviewer #3: No

---

## [Editor Report · Acceptance letter]

30 Oct 2024

PONE-D-24-28401R1 

PLOS ONE

Dear Dr. Minatoguchi, 

I'm pleased to inform you that your manuscript has been deemed suitable for publication in PLOS ONE. Congratulations! Your manuscript is now being handed over to our production team.

Kind regards, 

on behalf of

Dr. Wannaporn Ittiprasert 

Academic Editor

PLOS ONE